# A Generative Model for Electron Paths

**John Bradshaw**
University of Cambridge
Max Planck Institute, Tübingen
jab255@cam.ac.uk

**Matt J. Kusner**
University of Oxford
Alan Turing Institute
mkusner@turing.ac.uk

**Brooks Paige**
Alan Turing Institute
University of Cambridge
bpaige@turing.ac.uk

**Marwin H. S. Segler**
BenevolentAI
marwin.segler@benevolent.ai

**José Miguel Hernández-Lobato**
University of Cambridge
Microsoft Research Cambridge
Alan Turing Institute
jmh233@cam.ac.uk

## Abstract

Chemical reactions can be described as the stepwise redistribution of electrons in molecules. As such, reactions are often depicted using "arrow-pushing" diagrams which show this movement as a sequence of arrows. We propose an electron path prediction model (Electro) to learn these sequences directly from raw reaction data. Instead of predicting product molecules directly from reactant molecules in one shot, learning a model of electron movement has the benefits of (a) being easy for chemists to interpret, (b) incorporating constraints of chemistry, such as balanced atom counts before and after the reaction, and (c) naturally encoding the sparsity of chemical reactions, which usually involve changes in only a small number of atoms in the reactants. We design a method to extract approximate reaction paths from any dataset of atom-mapped reaction SMILES strings. Our model achieves excellent performance on an important subset of the USPTO reaction dataset, comparing favorably to the strongest baselines. Furthermore, we show that our model recovers a basic knowledge of chemistry without being explicitly trained to do so.

## 1 Introduction

The ability to reliably predict the products of chemical reactions is of central importance to the manufacture of medicines and materials, and to understand many processes in molecular biology. Theoretically, all chemical reactions can be described by the stepwise rearrangement of electrons in molecules (Herges, 1994b). This sequence of bond-making and breaking is known as the *reaction mechanism*. Understanding the reaction mechanism is crucial because it not only determines the products (formed at the last step of the mechanism), but it also provides insight into why the products are formed on an atomistic level. Mechanisms can be treated at different levels of abstraction. On the lowest level, quantum-mechanical simulations of the electronic structure can be performed, which are prohibitively computationally expensive for most systems of interest. On the other end, chemical reactions can be treated as rules that "rewrite" reactant molecules to products, which abstracts away the individual electron redistribution steps into a single, global transformation step. To combine the advantages of both approaches, chemists use a powerful qualitative model of quantum chemistry colloquially called "arrow pushing", which simplifies the stepwise electron shifts using sequences of arrows which indicate the path of electrons throughout molecular graphs (Herges, 1994b).

Recently, there have been a number of machine learning models proposed for directly predicting the products of chemical reactions (Coley et al., 2017; Jin et al., 2017; Schwaller et al., 2018; Segler and Waller, 2017a; Segler et al., 2018; Wei et al., 2016), largely using graph-based or machine translation models. The task of reaction product prediction is shown on the left-hand side of Figure 1.

In this paper we propose a machine learning model to predict the reaction mechanism, as shown on the right-hand side of Figure 1, for a particularly important subset of organic reactions. We argue that our

Figure 1: *(Left)* The reaction product prediction problem: Given the reactants and reagents, predict the structure of the product. *(Right)* The reaction mechanism prediction problem: Given the reactants and reagents, predict how the reaction occurred to form the products.

model is not only more interpretable than product prediction models, but also allows easier encoding of constraints imposed by chemistry. Proposed approaches to predicting reaction mechanisms have often been based on combining hand-coded heuristics and quantum mechanics (Bergeler et al., 2015; Kim et al., 2018; Nandi et al., 2017; Segler and Waller, 2017b; Rappoport et al., 2014; Simm and Reiher, 2017; Zimmerman, 2013), rather than using machine learning. We call our model ELECTRO, as it directly predicts the path of electrons through molecules (i.e., the reaction mechanism). To train the model we devise a general technique to obtain approximate reaction mechanisms purely from data about the reactants and products. This allows one to train our a model on large, unannotated reaction datasets such as USPTO (Lowe, 2012). We demonstrate that not only does our model achieve impressive results, surprisingly it also learns chemical properties it was not explicitly trained on.

## 2 BACKGROUND

We begin with a brief background from chemistry on molecules and chemical reactions, and then review related work in machine learning on predicting reaction outcomes. We then describe a particularly important subclass of chemical reactions, called *linear electron flow* (LEF) reactions, and summarize the contributions of this work.

### 2.1 MOLECULES AND CHEMICAL REACTIONS

Organic (carbon-based) molecules can be represented via a graph structure, where each node is an atom and each edge is a covalent bond (see example molecules in Figure 1). Each edge (bond) represents two electrons that are shared between the atoms that the bond connects.

Electrons are particularly important for describing how molecules react with other molecules to produce new ones. All chemical reactions involve the stepwise movement of electrons along the atoms in a set of reactant molecules. This movement causes the formation and breaking of chemical bonds that changes the reactants into a new set of product molecules (Herges, 1994a). For example, Figure 1 (*Right*) shows how electron movement can break bonds (red arrows) and make new bonds (green arrows) to produce a new set of product molecules.

### 2.2 RELATED WORK

In general, work in machine learning on reaction prediction can be divided into two categories: (1) *Product prediction*, where the goal is to predict the reaction products, given a set of reactants and reagents, shown in the left half of Figure 1; and (2) *Mechanism prediction*, where the goal is to determine *how* the reactants react, i.e., the movement of electrons, shown in the right of Figure 1.

**Product prediction.**    Recently, methods combining machine learning and template-based molecular rewriting rules have been proposed (Coley et al., 2017; Segler and Waller, 2017a; Segler et al., 2018; Wei et al., 2016; Zhang and Aires-de Sousa, 2005). Here, a learned model is used to predict which rewrite rule to apply to convert one molecule into another. While these models are readily interpretable, they tend be brittle. Another approach, introduced by Jin et al. (2017), constructs a neural network based on the Weisfeiler-Lehman algorithm for testing graph isomorphism. They use this algorithm (called WLDN) to select atoms that will be involved in a reaction. They then enumerate all chemically-valid bond changes involving these atoms and learn a separate network to

| Prior Work | end-to-end | mechanistic |
|---|---|---|
| Templates+ML | – | – |
| WLDN [Jin et al. (2017)] | – | – |
| Seq2Seq [Schwaller et al. (2018)] | ✓ | – |
| Source/Sink (expert-curated data) | – | ✓ |
| **This work** | ✓ | ✓ |

Table 1: Work on machine learning for reaction prediction, and whether they are (a) end-to-end trainable, and (b) predict the reaction mechanism.

rank the resulting potential products. This method, while leveraging new techniques for deep learning on graphs, cannot be trained end-to-end because of the enumeration steps for ensuring chemical validity. Schwaller et al. (2018) represents reactants as SMILES (Weininger, 1988) strings and then uses a sequence to sequence network (specifically, the work of Zhao et al. (2017)) to predict product SMILES. While this method (called Seq2Seq) is end-to-end trainable, the SMILES representation is quite brittle as often single character changes will not correspond to a valid molecule.

These latter two methods, WLDN and Seq2Seq, are state-of-the-art on product prediction and have been shown to outperform the above template-based techniques (Jin et al., 2017). Thus we compare directly with these two methods in this work.

**Mechanism prediction.** The only other work we are aware of to use machine learning to predict reaction mechanisms are Fooshee et al. (2018); Kayala and Baldi (2011; 2012); Kayala et al. (2011). All of these model a chemical reaction as an interaction between atoms as electron donors and as electron acceptors. They predict the reaction mechanisms via two independent models: one that identifies these likely electron sources and sinks, and another that ranks all combinations of them. These methods have been run on small expert-curated private datasets, which contain information about the reaction conditions such as the temperature and anion/cation solvation potential (Kayala and Baldi, 2011, §2). In contrast, in this work, we aim to learn reactions from noisy large-scale public reaction datasets, which are missing the required reaction condition information required by these previous works. As we cannot yet apply the above methods on the datasets we use, nor test our models on the datasets they use (as the data are not yet publicly released), we cannot compare directly against them; therefore, we leave a detailed investigation of the pros and cons of each method for future work.

As a whole, this related work points to at least two main desirable characteristics for reaction prediction models:

1. *End-to-End*: There are many complex chemical constraints that limit the space of all possible reactions. How can we differentiate through a model subject to these constraints?
2. *Mechanistic*: Learning the mechanism offers a number of benefits over learning the products directly including: interpretability (if the reaction failed, what electron step went wrong), sparsity (electron steps only involve a handful of atoms), and generalization (unseen reactions also follow a set of electron steps).

Table 1 describes how the current work on reaction prediction satisfies these characteristics. In this work we propose to model a subset of mechanisms with *linear electron flow*, described below.

## 2.3 LINEAR ELECTRON FLOW REACTIONS

Reaction mechanisms can be classified by the topology of their "electron-pushing arrows" (the red and green arrows in Figure 1). Here, the class of reactions with *linear electron flow* (LEF) topology is by far the most common and fundamental, followed by those with cyclic topology (Herges, 1994a). In this work, we will only consider LEF reactions that are *heterolytic*, i.e., they involve pairs of electrons.[1]

---

[1] The treatment of radical reactions, which involve the movement of single electrons, will be the topic of future work.

If reactions fall into this class, then a chemical reaction can be modelled as pairs of electrons moving in a *single path* through the reactant atoms. In arrow pushing diagrams representing LEF reactions, this electron path can be represented by arrows that line up in sequence, differing from for example pericyclic reactions in which the arrows would form a loop (Herges, 1994a).

Further for LEF reactions, the movement of the electrons along the linear path will alternately remove existing bonds and form new ones. We show this alternating structure in the right of Figure 1. The reaction formally starts by (step 1) taking the pair of electrons between the Li and C atoms and moving them to the C atom; this is a remove bond step. Next (step 2) a bond is added when electrons are moved from the C atom in reactant 1 to a C atom in reactant 2. Then (step 3) a pair of electrons are removed between the C and O atoms and moved to the O atom, giving rise to the products. Predicting the final product is thus a byproduct of predicting this series of electron steps.

**Contributions.**  We propose a novel generative model for modeling the reaction mechanism of LEF reactions. Our contributions are as follows:

- We propose an end-to-end generative model for predicting reaction mechanisms, ELECTRO, that is fully differentiable. It can be used with any deep learning architecture on graphs.
- We design a technique to identify LEF reactions and mechanisms from purely atom-mapped reactants and products, the primary format of large-scale reaction datasets.
- We show that ELECTRO learns chemical knowledge such as functional group selectivity without explicit training.

## 3   THE GENERATIVE MODEL

In this section we define a probabilistic model for electron movement in *linear electron flow* (LEF) reactions. As described above (§2.1) all molecules can be thought of as graphs where nodes correspond to atoms and edges to bonds. All LEF reactions transform a set of reactant graphs, $\mathcal{M}_0$ into a set of product graphs $\mathcal{M}_{T+1}$ via a series of electron actions $\mathcal{P}_{0:T} = (a_0, \ldots, a_T)$. As described, these electron actions will alternately *remove* and *add* bonds (as shown in the right of Figure 1). This reaction sometimes includes additional reagent graphs, $\mathcal{M}_e$, which help the reaction proceed, but do not change themselves. We propose to learn a distribution $p_\theta(\mathcal{P}_{0:T} \mid \mathcal{M}_0, \mathcal{M}_e)$ over these electron movements. We first detail the generative process that specifies $p_\theta$, before describing how to train the model parameters $\theta$.

To define our generative model, we describe a factorization of $p_\theta(\mathcal{P}_{0:T} \mid \mathcal{M}_0, \mathcal{M}_e)$ into three components: 1. the starting location distribution $p_\theta^{\text{start}}(a_0 \mid \mathcal{M}_0, \mathcal{M}_e)$; 2. the electron movement distribution $p_\theta(a_t \mid \mathcal{M}_t, a_{t-1}, t)$; and 3. the reaction continuation distribution $p_\theta^{\text{cont}}(c_t \mid \mathcal{M}_t)$. We define each of these in turn and then describe the factorization (we leave all architectural details of the functions introduced to the appendix).

**Starting Location.**  At the beginning the model needs to decide on which atom $a_0$ starts the path. As this is based on (i) the initial set of reactants $\mathcal{M}_0$ and possibly (ii) a set of reagents $\mathcal{M}_e$, we propose to learn a distribution $p_\theta^{\text{start}}(a_0 \mid \mathcal{M}_0, \mathcal{M}_e)$.

To parameterize this distribution we propose to use any deep graph neural network, denoted $h_{\mathcal{A}}(\cdot)$, to learn graph-isomorphic node features from the initial atom and bond features[2] (Duvenaud et al., 2015; Kipf and Welling, 2017; Li et al., 2016; Gilmer et al., 2017). We choose to use a 4 layer Gated Graph Neural Network (GGNN) (Li et al., 2016), for which we include a short review in the appendix.

Given these atom embeddings we also compute graph embeddings (Li et al., 2018, §B.1) (also called an aggregation graph transformation (Johnson, 2017, §3)), which is a vector that represents the entire molecule set $\mathcal{M}$ that is invariant to any particular node ordering. Any such function $g(\cdot)$ that computes this mapping can be used here, but the particular graph embedding function we use is inspired by Li et al. (2018), and described in detail in Appendix B. We can now parameterize $p_\theta^{\text{start}}(a_0 \mid \mathcal{M}_0, \mathcal{M}_e)$ as

$$p_\theta^{\text{start}}(a_0 \mid \mathcal{M}_0, \mathcal{M}_e) = \text{softmax}\Big[ f^{\text{start}}\big( h_{\mathcal{A}}(\mathcal{M}_0), g^{\text{reagent}}\left(\mathcal{M}_e\right) \big) \Big], \tag{1}$$

---

[2]The molecular features we use are described in Table 4 in Appendix B.

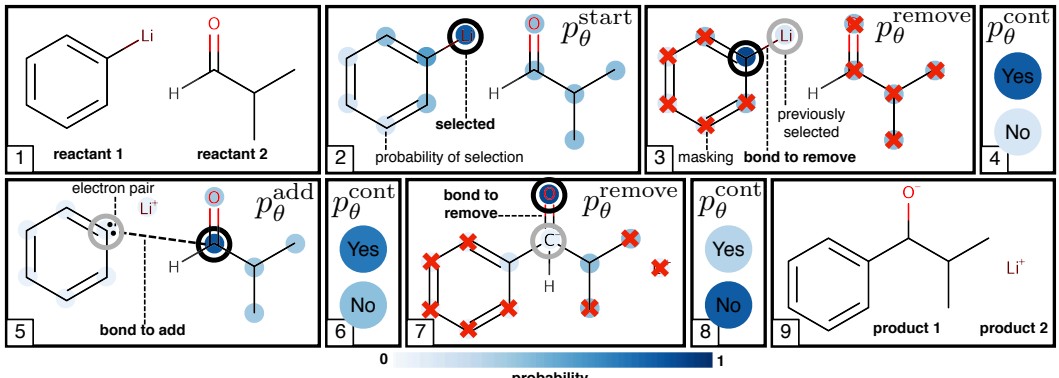

Figure 2: This figure shows the sequence of actions in transforming the reactants in box 1 to the products in box 9. The sequence of actions will result in a sequence of pairs of atoms, between which bonds will alternately be removed and created, creating a series of intermediate products. At each step the model sees the current intermediate product graph (shown in the boxes) as well as the previous action, if applicable, shown by the grey circle. It uses this to decide on the next action. We represent the characteristic probabilities the model may have over these next actions as colored circles over each atom. Some actions are disallowed on certain steps, for instance you cannot remove a bond that does not exist; these blocked actions are shown as red crosses.

where $f^{\text{start}}$ is a feedforward neural network which computes logits $\mathbf{x}$; the logits are then normalized into probabilities by the softmax function, defined as $\text{softmax}[\mathbf{x}] = e^{\mathbf{x}} / \sum_i e^{x_i}$.

**Electron Movement.** Observe that since LEF reactions are a single path of electrons (§2.3), at any step $t$, the next step $a_t$ in the path depends only on (i) the intermediate molecules formed by the action path up to that point $\mathcal{M}_t$, (ii) the previous action taken $a_{t-1}$ (indicating where the free pair of electrons are), and (iii) the point of time $t$ through the path, indicating whether we are on an add or remove bond step. Thus we will also learn the electron movement distribution $p_\theta(a_t \mid \mathcal{M}_t, a_{t-1}, t)$.

Similar to the starting location distribution we again make use of a graph-isomorphic node embedding function $h_{\mathcal{A}}(\mathcal{M})$. In contrast, the above distribution can be split into two distributions depending on the parity of $t$: the remove bond step distribution $p_\theta^{\text{remove}}(a_t \mid \mathcal{M}_t, a_{t-1})$ when $t$ is odd, and the add bond step distribution $p_\theta^{\text{add}}(a_t \mid \mathcal{M}_t, a_{t-1})$ when $t$ is even. We parameterize the distributions as

$$p_\theta^{\text{remove}}(a_t \mid \mathcal{M}_t, a_{t-1}) \propto \boldsymbol{\beta}^{\text{remove}} \odot \text{softmax}\Big[ f^{\text{remove}}\big(h_{\mathcal{A}}(\mathcal{M}_t), a_{t-1}\big)\Big], \tag{2}$$

$$p_\theta^{\text{add}}(a_t \mid \mathcal{M}_t, a_{t-1}) \propto \boldsymbol{\beta}^{\text{add}} \odot \text{softmax}\Big[ f^{\text{add}}\big(h_{\mathcal{A}}(\mathcal{M}_t), a_{t-1}\big)\Big] \tag{3}$$

$$p_\theta(a_t \mid \mathcal{M}_t, a_{t-1}, t) = \begin{cases} p_\theta^{\text{remove}}(a_t \mid \mathcal{M}_t, a_{t-1}) & \text{if } t \text{ is odd} \\ p_\theta^{\text{add}}(a_t \mid \mathcal{M}_t, a_{t-1}) & \text{otherwise} \end{cases} \tag{4}$$

The vectors $\boldsymbol{\beta}^{\text{remove}}, \boldsymbol{\beta}^{\text{add}}$ are masks that zero-out the probability of certain atoms being selected. Specifically, $\boldsymbol{\beta}^{\text{remove}}$ sets the probability of any atoms $a_t$ to 0 if there is not a bond between it and the previous atom $a_{t-1}$[3]. The other mask vector $\boldsymbol{\beta}^{\text{add}}$ masks out the previous action, preventing the model from stalling in the same state for multiple time-steps. The feedforward networks $f^{\text{add}}(\cdot), f^{\text{remove}}(\cdot)$ and other architectural details are described in Appendix C.

**Reaction Continuation / Termination.** Additionally, as we do not know the length of the reaction $T$, we introduce a latent variable $c_t \in \{0, 1\}$ at each step $t$, which describes whether the reaction continues ($c_t = 1$) or terminates ($c_t = 0$) [4]. We also define an upper bound $T^{\text{max}}$ on the number of reaction steps.

---

[3]One subtle point is if a reaction begins with a lone-pair of electrons then we say that this reaction starts by removing a *self-bond*. Thus, in the first remove step $\boldsymbol{\beta}^{\text{remove}}$ it is possible to select $a_1 = a_0$. But this is not allowed via the mask vector in later steps.

[4]An additional subtle point is that we do not allow the reaction to stop until until it has picked up an entire pair (ie $c_1 = 1$).

---

**Algorithm 1** The generative steps of ELECTRO (given that the model chooses to react, ie $c_0 = 1$).

---

**Input:** Reactant molecules $\mathcal{M}_0$ (consisting of atoms $\mathcal{A}$), reagents $\mathcal{M}_e$, atom embedding function $h_{\mathcal{A}}(\cdot)$, graph embedding functions $g^{\text{reagent}}(\cdot)$ and $g^{\text{cont}}(\cdot)$, additional logit functions $f^{\text{start}}(\cdot), f^{\text{remove}}(\cdot), f^{\text{add}}(\cdot)$, time steps $T^{\text{max}}$

1:   $p_\theta^{\text{start}}(a \mid \mathcal{M}_0, \mathcal{M}_e) \triangleq \text{softmax}\left[f^{\text{start}}\left(h_{\mathcal{A}}(\mathcal{M}_0), g^{\text{reagent}}(\mathcal{M}_e)\right)\right]$    ▷*a starts reaction*

2:   $a_0 \sim p_\theta^{\text{start}}(a \mid \mathcal{M}_0, \mathcal{M}_e)$

3:   $\mathcal{M}_1 \leftarrow \mathcal{M}_0$    ▷*The molecule does not change until complete pair picked up*

4:   $c_1 \triangleq 1$    ▷*You cannot stop until picked up complete pair*

5:   **for** $t = 1, \ldots, T^{\text{max}}$ **do**

6:     **if** $t$ is odd **then**

7:       $p_\theta^{\text{remove}}(a_t|\mathcal{M}_t, a_{t-1}) \propto \boldsymbol{\beta}^{\text{remove}}\text{softmax}\left[f^{\text{remove}}\left(h_{\mathcal{A}}(\mathcal{M}_t), a_{t-1}\right)\right]$

8:       $a_t \sim p_\theta^{\text{remove}}(a_t|\mathcal{M}_t, a_{t-1})$    ▷*electrons remove bond between $a_t$ and $a_{t-1}$*

9:     **else**

10:      $p_\theta^{\text{add}}(a_t|\mathcal{M}_t, a_{t-1}) \propto \boldsymbol{\beta}^{\text{add}}\text{softmax}\left[f^{\text{add}}\left(h_{\mathcal{A}}(\mathcal{M}_t), a_{t-1}\right)\right]$

11:      $a_t \sim p_\theta^{\text{add}}(a_t|\mathcal{M}_t, a_{t-1})$    ▷*electrons add bond between $a_t$ and $a_{t-1}$*

12:     **end if**

13:     $\mathcal{P}_t = \mathcal{P}_{0:t-1} \cup a_t$

14:     $\mathcal{M}_{t+1} \leftarrow \mathcal{M}_t, a_t$    ▷*modify molecules based on previous molecule and action*

15:     $p_\theta^{\text{cont}}(c_{t+1} \mid \mathcal{M}_{t+1}) \triangleq \sigma(g^{\text{cont}}(\mathcal{M}_{t+1}))$

16:     $c_{t+1} \sim p_\theta^{\text{cont}}(c_{t+1} \mid \mathcal{M}_{t+1})$    ▷*whether to continue reaction*

17:     **if** $c_{t+1} = 0$ **then**

18:       break

19:     **end if**

20:  **end for**

**Output:** Electron path $\mathcal{P}_{0:t}$

---

The final distribution we learn is the continuation distribution $p_\theta^{\text{cont}}(c_t \mid \mathcal{M}_t)$. For this distribution we learn a different graph embedding function $g^{\text{cont}}(\cdot)$ to decide whether to continue or not:

$$p_\theta^{\text{cont}}(c_t \mid \mathcal{M}_t) = \sigma(g^{\text{cont}}(\mathcal{M}_t)). \tag{5}$$

where $\sigma$ is the sigmoid function $\sigma(a) = 1/(1 + e^{-a})$.

**Path Distribution Factorization.** Given these distributions we can define the probability of a path $\mathcal{P}_{0:T}$ with the distribution $p_\theta(\mathcal{P}_{0:T} \mid \mathcal{M}_0, \mathcal{M}_e)$, which factorizes as

$$p_\theta(\mathcal{P}_{0:T} \mid \mathcal{M}_0, \mathcal{M}_e) = p_\theta^{\text{cont}}(c_0 \mid \mathcal{M}_0)p_\theta^{\text{start}}(a_0 \mid \mathcal{M}_0, \mathcal{M}_e) \tag{6}$$

$$\times \left[\prod_{t=1}^{T} p_\theta^{\text{cont}}(c_t \mid \mathcal{M}_t)p_\theta(a_t \mid \mathcal{M}_t, a_{t-1}, t)\right]\left(1 - p_\theta^{\text{cont}}(c_{T+1} \mid \mathcal{M}_{T+1})\right),$$

Figure 2 gives a graphical depiction of the generative process on a simple example reaction. Algorithm 1 gives a more detailed description.

**Training** We can learn the parameters $\theta$ of all the parameterized functions, including those producing node embeddings, by maximizing the log likelihood of a full path $\log p_\theta(\mathcal{P}_{0:T} \mid \mathcal{M}_0, \mathcal{M}_e)$. This is evaluated by using a known electron path $a_t^\star$ and intermediate products $\mathcal{M}_t^\star$ extracted from training data, rather than on simulated values. This allows us to train on all stages of the reaction at once, given electron path data. We train our models using Adam (Kingma and Ba, 2015) and an initial learning rate of $10^{-4}$, with minibatches consisting of a single reaction, where each reaction often consists of multiple intermediate graphs.

**Prediction** Once trained, we can use our model to sample chemically-valid paths given an input set of reactants $\mathcal{M}_0$ and reagents $\mathcal{M}_e$, simply by simulating from the conditional distributions until sampling a continue value equal to zero. We instead would like to find a ranked list of the top-$K$ predicted paths, and do so using a modified beam search, in which we roll out a beam of width $K$ until a maximum path length $T^{\text{max}}$, while recording all paths which have terminated. This search procedure is described in detail in Algorithm 2 in the appendix.

## 4 REACTION MECHANISM IDENTIFICATION

To evaluate our model, we use a collection of chemical reactions extracted from the US patent database (Lowe, 2017). We take as our starting point the 479,035 reactions, along with the training, validation, and testing splits, which were used by Jin et al. (2017), referred to as the USPTO dataset. This data consists of a list of reactions. Each reaction is a reaction SMILES string (Weininger, 1988) and a list of bond changes. SMILES is a text format for molecules that lists the molecule as a sequence of atoms and bonds. The bond change list tells us which pairs of atoms have different bonds in the the reactants versus the products (note that this can be directly determined from the SMILES string). Below, we describe two data processing techniques that allow us to identify reagents, reactions with LEF topology, and extract an underlying electron path. Each of these steps can be easily implemented with the open-source chemo-informatics software RDKit (RDKit, online).

**Reactant and Reagent Seperation** Reaction SMILES strings can be split into three parts — reactants, reagents, and products. The reactant molecules are those which are consumed during the course of the chemical reaction to form the product, while the *reagents* are any additional molecules which provide context under which the reaction occurs (for example, catalysts), but do not explicitly take part in the reaction itself; an example of a reagent is shown in Figure 1.

Unfortunately, the USPTO dataset as extracted does not differentiate between reagents and reactants. We elect to preprocess the entire USPTO dataset by separating out the reagents from the reactants using the process outlined in Schwaller et al. (2018), where we classify as a reagent any molecule for which either (i) none of its constituent atoms appear in the product, or (ii) the molecule appears in the product SMILES completely unchanged from the pre-reaction SMILES. This allows us to properly model molecules which are included in the dataset but do not materially contribute to the reaction.

**Identifying Reactions with Linear Electron Flow Topology** To train our model, we need to (i) identify reactions in the USPTO dataset with LEF topology, and (ii) have access to an electron path for each reaction. Figure 3 shows the steps necessary to identify and extract the electron paths from reactions exhibiting LEF topology. We provide further details in Appendix D.

Applying these steps, we discover that $73\%$ of the USPTO dataset consists of LEF reactions (349,898 total reactions, of which 29,360 form the held-out test set).

## 5 EXPERIMENTS AND EVALUATION

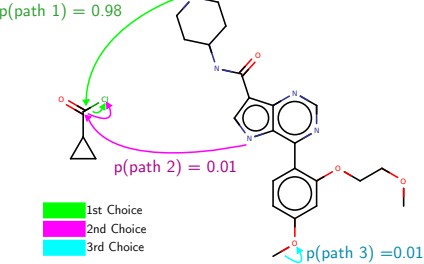

Figure 4: An example of the paths suggested by ELECTRO-LITE on one of the USPTO test examples. Its first choice in this instance is correct.

| | Accuracies (%) | | | |
|---|---|---|---|---|
| Model Name | Top-1 | Top-2 | Top-3 | Top-5 |
| ELECTRO-LITE | 70.3 | 82.8 | 87.7 | 92.2 |
| ELECTRO | 77.8 | 89.2 | 92.4 | 94.7 |

Table 2: Results when using ELECTRO for *mechanism prediction*. Here a prediction is correct if the atom mapped action sequences predicted by our model match exactly those extracted from the USPTO dataset.

We now evaluate ELECTRO on the task of (i) *mechanism prediction* and (ii) *product prediction* (as described in Figure 1). While generally, it is necessary to know the reagents $\mathcal{M}_e$ of a reaction to faithfully predict the mechanism and product, it is often possible to make inferences from the reactants alone. Therefore, we trained a second version of our model that we call ELECTRO-LITE, which ignores reagent information. This allows us to gauge the importance of reagents in determining the mechanism of the reaction.

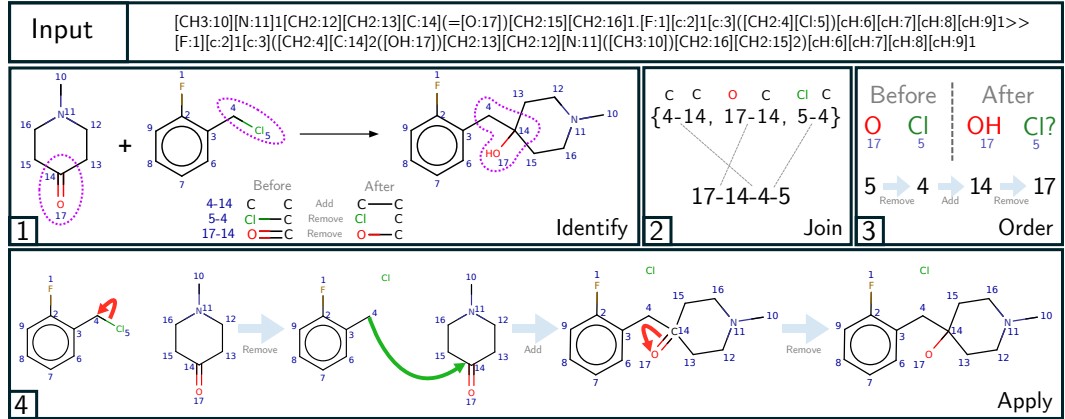

Figure 3: Example of how we turn a SMILES reaction string into an ordered electron path, for which we can train ELECTRO on. This consists of a series of steps: (1) Identify bonds that change by comparing bond triples (source node, end node, bond type) between the reactants and products. (2) Join up the bond changes so that one of the atoms in consecutive bond changes overlap (for reactions which do not have linear electron flow topology, such as multi-step reactions, this will not be possible and so we discard these reactions). (3) Order the path (ie assign a direction). A gain of charge (or analogously the gain of hydrogen as $H^+$ ions without changing charge, such as in the example shown) indicates that the electrons have arrived at this atom; and vice-versa for the start of the path. When details about both ends of the path are missing from the SMILES string we fall back to using an element's *electronegativity* to estimate the direction of our path, with more electronegative atoms attracting electrons towards them and so being at the end of the path. (4) The extracted electron path deterministically determines a series of intermediate molecules which can be used for training ELECTRO. Paths that do not consist of alternative add and removal steps and do not result in the final recorded product do not exhibit LEF topology and so can be discarded. An interesting observation is that our approximate reaction mechanism extraction scheme implicitly fills in missing reagents, which are caused by noisy training data — in this example, which is a Grignard- or Barbier-type reaction, the test example is missing a metal reagent (e.g. Mg or Zn). Nevertheless, our model is robust enough to predict the intended product correctly (Effland et al., 1981).

## 5.1 REACTION MECHANISM PREDICTION

For mechanism prediction we are interested in ensuring we obtain the exact sequence of electron steps correctly. We evaluate accuracy by checking whether the sequence of integers extracted from the raw data as described in Section 4 is an exact match with the sequence of integers output by ELECTRO. We compute the top-1, top-2, top-3, and top-5 accuracies and show them in Table 2, with an example prediction shown in Figure 4.

## 5.2 REACTION PRODUCT PREDICTION

Reaction mechanism prediction is useful to ensure we form the correct product in the *correct way*. However, it underestimates the model's actual predictive accuracy: although a single atom mapping is provided as part of the USPTO dataset, in general atom mappings are not unique (e.g., if a molecule contains symmetries). Specifically, multiple different sequences of integers could correspond to chemically-identical electron paths. The first figure in the appendix shows an example of a reaction with symmetries, where different electron paths produce the exact same product.

Recent approaches to *product prediction* (Jin et al., 2017; Schwaller et al., 2018) have evaluated whether the major product reported in the test dataset matches predicted candidate products generated by their system, independent of mechanism. In our case, the top-5 accuracy for a particular reaction may include multiple different electron paths that ultimately yield the same product molecule.

To evaluate if our model predicts the same major product as the one in the test data, we need to solve a graph isomorphism problem. To approximate this we (a) take the predicted electron path, (b) apply these edits to the reactants to produce a product graph (balancing charge to satisfy valence

|  | Accuracies (%) | | | |
| Model Name | Top-1 | Top-2 | Top-3 | Top-5 |
| --- | --- | --- | --- | --- |
| WLDN FTS (Jin et al., 2017) | 84.0 | 89.2 | 91.1 | 92.3 |
| WLDN (Jin et al., 2017) | 83.1 | 89.3 | 91.5 | 92.7 |
| Seq2Seq FTS (Schwaller et al., 2018) | 81.7 | 86.8 | 88.4 | 89.8 |
| Seq2Seq (Schwaller et al., 2018) | 82.6 | 87.3 | 88.8 | 90.1 |
| ELECTRO-LITE | 78.2 | 87.7 | 91.5 | 94.4 |
| ELECTRO | **87.0** | **92.6** | **94.5** | **95.9** |

Table 3: Results for *product prediction*, following the product matching procedure in Section 5.2. For the baselines we compare against models trained (a) on the full USPTO training set (marked FTS) and only tested on our subset of LEF reactions, and (b) those that are also trained on the same subset as our model. We make use of the code and pre-trained models provided by Jin et al. (2017). For the Seq2Seq approach, as neither code nor more fine grained results are available, we train up the required models from scratch using the OpenNMT library (Klein et al., 2017).

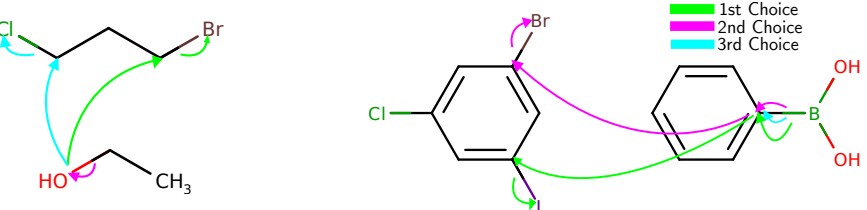

Figure 5: (Left) Nucleophilic substitutions $S_N 2$-reactions, (right) Suzuki-coupling (note that in the "real" mechanism of the Suzuki coupling, the reaction would proceed via oxidative insertion, transmetallation and reductive elimination at a Palladium catalyst. As these details are not contained in training data, we treat Palladium implicitly as a reagent). In both cases, our model has correctly picked up the trend that halides lower in the period table usually react preferably ($I > Br > Cl$).

constraints), (c) remove atom mappings, and (d) convert the product graph to a canonical SMILES string representation in Kekulé form (aromatic bonds are explicitly represented as double-bonds). We can then evaluate whether a predicted electron path matches the ground truth by a string comparison. This procedure is inspired by the evaluation of Schwaller et al. (2018). To obtain a ranked list of products for our model, we compute this canonicalized product SMILES for each of the predictions found by beam search over electron paths, removing duplicates along the way. These product-level accuracies are reported in Table 3.

We compare with the state-of-the-art graph-based method Jin et al. (2017); we use their evaluation code and pre-trained model[5], re-evaluated on our extracted test set. We also use their code and re-train a model on our extracted training set, to ensure that any differences between our method and theirs is not due to a specialized training task. We also compare against the Seq2Seq model proposed by (Schwaller et al., 2018); however, as no code is provided by Schwaller et al. (2018), we run our own implementation of this method based on the OpenNMT library (Klein et al., 2017). Overall, ELECTRO outperforms all other approaches on this task, with 87% top-1 accuracy and 95.9% top-5 accuracy. Omitting the reagents in ELECTRO degrades top-1 accuracy slightly, but maintains a high top-3 and top-5 accuracy, suggesting that reagent information is necessary to provide context in disambiguating plausible reaction paths.

## 5.3 QUALITATIVE ANALYSIS

Complex molecules often feature several potentially reactive functional groups, which compete for reaction partners. To predict the selectivity, that is which functional group will predominantly react in the presence of other groups, students of chemistry learn heuristics and trends, which have been established over the course of three centuries of experimental observation. To qualitatively study

---

[5]https://github.com/wengong-jin/nips17-rexgen

whether the model has learned such trends from data we queried the model with several typical text book examples from the chemical curriculum (see Figure 5 and the appendix). We found that the model predicts most examples correctly. In the few incorrect cases, interpreting the model's output reveals that the model made chemically plausible predictions.

# 6 LIMITATIONS AND FUTURE DIRECTIONS

In this section we briefly list a couple of limitations of our approach and discuss any pointers towards their resolution in future work.

**LEF Topology** ELECTRO can currently only predict reactions with LEF topology (§2.3). These are the most common form of reactions (Herges, 1994b), but in future work we would like to extend ELECTRO's action repertoire to work with other classes of electron shift topologies such as those found in pericyclic reactions. This could be done by allowing ELECTRO to sequentially output a series of paths, or by allowing multiple electron movements at a single step. Also, since the approximate mechanisms we produce for our dataset are extracted only from the reactants and products, they may not include all observable intermediates. This could be solved by using labelled mechanism paths, obtainable from finer grained datasets containing also the mechanistic intermediates. These mechanistic intermediates could also perhaps be created using quantum mechanical calculations following the approach in Sadowski et al. (2016).

**Graph Representation of Molecules** Although this shortcoming is not just restricted to our work, by modeling molecules and reactions as graphs and operations thereon, we ignore details about the electronic structure and conformational information, ie information about how the atoms in the molecule are oriented in 3D. This information is crucial in some important cases. Having said this, there is probably some balance to be struck here, as representing molecules and reactions as graphs is an extremely powerful abstraction, and one that is commonly used by chemists, allowing models working with such graph representations to be more easily interpreted.

# 7 CONCLUSION

In this paper we proposed ELECTRO, a model for predicting electron paths for reactions with linear electron flow. These electron paths, or *reaction mechanisms*, describe how molecules react together. Our model (i) produces output that is easy for chemists to interpret, and (ii) exploits the sparsity and compositionality involved in chemical reactions. As a byproduct of predicting reaction mechanisms we are also able to perform reaction product prediction, comparing favorably to the strongest baselines on this task.

## ACKNOWLEDGEMENTS

We would like to thank Jennifer Wei, Dennis Sheberla, and David Duvenaud for their very helpful discussions. This work was supported by The Alan Turing Institute under the EPSRC grant EP/N510129/1. JB also acknowledges support from an EPSRC studentship.

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

## A  EXAMPLE OF SYMMETRY AFFECTING EVALUATION OF ELECTRON PATHS

In the main text we described the challenges of how to evaluate our model, as different electron paths can form the same products, for instance due to symmetry. Figure 6 is an example of this.

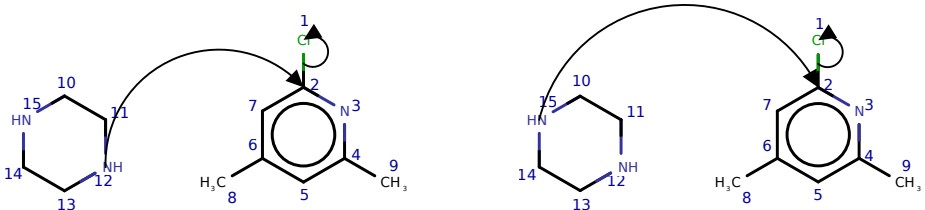

(a) Reaction as defined by USPTO SMILES

(b) Possible action sequences that all result in same major product.

Figure 6: This example shows how symmetry can affect the evaluation of electron paths. In this example, although one electron path is given in the USPTO dataset, the initial N that reacts could be either 15 or 12, with no difference in the final product. This is why judging purely based on electron path accuracy can sometimes be misleading.

## B  FORMING NODE AND GRAPH EMBEDDINGS

In this section we briefly review existing work for forming node and graph embeddings, as well as describing more specific details relating to our particular implementation of these methods. Figure 7 provides a visualization of these techniques. We follow the main text by denoting a set of molecules as $\mathcal{M}$, and refer to the atoms in these molecules (which are represented as nodes in a graph) as $\mathcal{A}$.

We start with Gated Graph Neural Networks (GGNNs) (Li et al., 2016; Gilmer et al., 2017), which we use for finding node embeddings. We denote these functions as $h_{\mathcal{A}} : \mathcal{M} \to \mathbb{R}^{|\mathcal{A}| \times d}$, where we will refer to the output as the node embedding matrix, $\mathbf{H}_{\mathcal{M}} \in \mathbb{R}^{|\mathcal{A}| \times d}$. Each row of this node embedding matrix represents the embedding of a particular atom; the rows are ordered by atom-mapped number, a unique number assigned to each atom in a SMILES string. The GGNN form these node embeddings through a recurrent operation on messages, $\mathbf{m}_v$, with $v \in \mathcal{A}$, so that there is one message associated with each node. At the first time step these messages, $\mathbf{m}_v^{(0)}$, are initialized with the respective atom features shown in Table 4. GGNNs then update these messages in a recursive nature:

$$\mathbf{m}_v^{(s)} = \text{GRU}\left(\mathbf{m}_v^{(s-1)}, \sum_{i \in \mathcal{N}_{e1}(v)} f_{\text{single}}\left(\mathbf{m}_i^{(s-1)}\right) + \sum_{j \in \mathcal{N}_{e2}(v)} f_{\text{double}}\left(\mathbf{m}_j^{(s-1)}\right) + \sum_{k \in \mathcal{N}_{e3}(v)} f_{\text{triple}}\left(\mathbf{m}_k^{(s-1)}\right)\right)$$
(7)

Where GRU is a Gated Recurrent Unit (Cho et al., 2014), the functions $\mathcal{N}_{e1}(v)$, $\mathcal{N}_{e2}(v)$, $\mathcal{N}_{e3}(v)$ index the nodes connected by single, double and triple bonds to node $v$ respectively and $f_{\text{single}}(\cdot)$, $f_{\text{double}}(\cdot)$ and $f_{\text{triple}}(\cdot)$ are linear transformations with learnable parameters. This process continues for $S$ steps (where we choose $S = 4$). In our implementation, messages and the hidden layer of the GRU have a dimensionality of 101, which is the same as the dimension of the raw atom features. The node embeddings are set as the final message belonging to a node, so that indexing a row of the node embeddings matrix, $\mathbf{H}_{\mathcal{M}}$, gives a transpose of the final message vector, ie $[\mathbf{H}_{\mathcal{M}}]_v = \mathbf{m}_v^{(S)t}$.

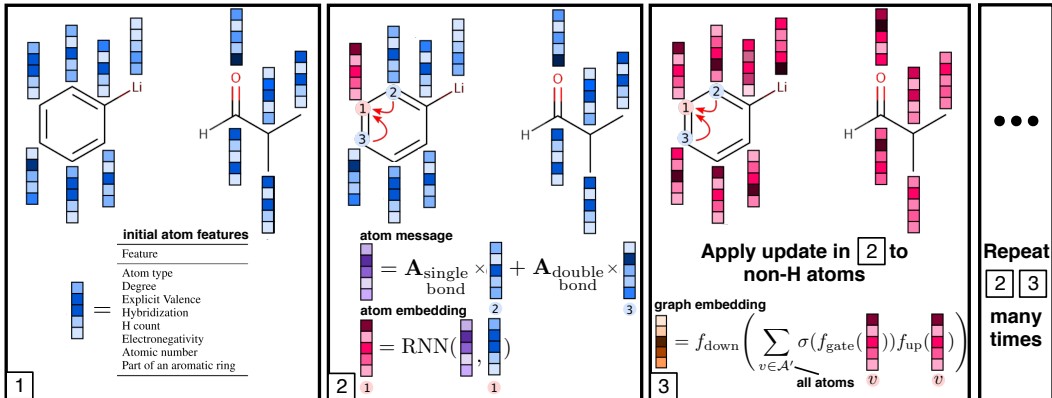

Figure 7: Visualization of how node embeddings and graph embeddings are formed. Node embeddings are $d$-dimensional vectors, one for each node. They are obtained using Gated Graph Neural Networks (Li et al., 2016). These networks consist of a series of iterative steps where the embeddings for each node are updated using the node's previous embedding and a message from its neighbors. Graph embeddings are $q$-dimensional vectors, representing a set of nodes, which could for instance be all the nodes in a particular graph (Li et al., 2018). They are formed using a function on the weighted sum of node embeddings.

Table 4: Atom features we use as input to the GGNN. These are calculated using RDKit.

| Feature | Description |
|---|---|
| Atom type | 72 possible elements in total, one hot |
| Degree | One hot (0, 1, 2, 3, 4, 5, 6, 7, 10) |
| Explicit Valence | One hot (0, 1, 2, 3, 4, 5, 6, 7, 8, 10, 12, 14) |
| Hybridization | One hot (SP, SP2, SP3, Other) |
| H count | integer |
| Electronegativity | float |
| Atomic number | integer |
| Part of an aromatic ring | boolean |

One can represent entire graphs with graph embeddings (Li et al., 2018; Johnson, 2017), which are $q$-dimensional vectors representing a set of nodes; i.e. an entire molecule or set of molecules. These are computed by the function $g : \mathcal{M} \to \mathbb{R}^q$. In practice the function we use consists of the composition of two functions: $g(\cdot) = (r \circ h_{\mathcal{A}})(\cdot)$.

Having already introduced the function $h_{\mathcal{A}}(\cdot)$, we now introduce the function $r(\cdot)$. This function, that maps a set of node features to graph embeddings, $r : \mathbb{R}^{|\mathcal{A}| \times d} \to \mathbb{R}^q$, is similar to the readout functions used for regressing on graphs detailed in (Gilmer et al., 2017, Eq. 3) and the graph embeddings described in Li et al. (2018, §B.1). Specifically, $r(\cdot)$ consists of three functions, $f_{\text{gate}}(\cdot)$, $f_{\text{up}}(\cdot)$ and $f_{\text{down}}(\cdot)$, which could be any multi-layer perceptron (MLP) but in practice we find that linear functions suffice. These three functions are used to form the graph embedding as so:

$$r(\mathbf{H}_{\mathcal{M}_t}) = f_{\text{down}} \left( \sum_{v \in \mathcal{A}'} \sigma \left( f_{\text{gate}}([\mathbf{H}_{\mathcal{M}}]_v) \right) f_{\text{up}}([\mathbf{H}_{\mathcal{M}}]_v) \right). \tag{8}$$

Where $\sigma(\cdot)$ is a sigmoid function. We can break this equation down into two stages. In stage (i), similar to Li et al. (2018, §B.1), we form an embedding of one or more molecules (with vertices $\mathcal{A}'$ and with $\mathcal{A}' \subseteq \mathcal{A}$) by performing a gated sum over the node features. In this manner the function $f_{\text{gate}}(\cdot)$ is used to decide how much that node should contribute towards the embedding, and $f_{\text{up}}(\cdot)$ projects the node embedding up to a higher dimensional space; following Li et al. (2018, §B.1), we choose this to be double the dimension of the node features. Having formed this embedding of the graphs, we project this down to a lower $q$-dimensional space in stage (ii), which is done by the function $f_{\text{down}}(\cdot)$.

## C    MORE TRAINING DETAILS

In this section we go through more specific model architecture and training details omitted from the main text.

### C.1    MODEL ARCHITECTURES

In this section we provide further details of our model architectures.

Section 3 of the main paper discusses our model. In particular we are interested in computing three conditional probability terms: (1) $p_{\theta}^{\text{start}}(a_0 \mid \mathcal{M}_0, \mathcal{M}_e)$, the probability of the initial state $a_0$ given the reactants and reagents; (2) the conditional probability $p_{\theta}(a_t \mid \mathcal{M}_t, a_{t-1}, t)$ of the next state $a_t$ given the intermediate products $\mathcal{M}_t$ for $t > 0$; and (3) the probability $p_{\theta}^{\text{cont}}(c_t \mid \mathcal{M}_t)$ that the reaction continues at step $t$.

Each of these is parametrized by NNs. We can split up the components of these NNs into a series of modules: $r^{\text{cont}}(\cdot)$, $r^{\text{reagent}}(\cdot)$, $f^{\text{add}}(\cdot)$, $f^{\text{remove}}(\cdot)$ and $f^{\text{start}}(\cdot)$. All of these operate on node embeddings created by the same GGNN. In this section we shall go through each of these modules in turn.

As mentioned above (Eq. 8) both $r^{\text{cont}}(\cdot)$ and $r^{\text{reagent}}(\cdot)$ (which following the explanation in the previous section, make up part of $g^{\text{cont}}(\cdot)$ and $g^{\text{reagent}}(\cdot)$ respectively) consist of three linear functions. For both, the function $f_{\text{gate}}(\cdot)$ is used to decide how much each node should contribute towards the embedding and so projects down to a scalar value. Again for both, $f_{\text{up}}(\cdot)$ projects the node embedding up to a higher dimensional space, which we choose to be 202 dimensions. This is double the dimension of the node features, and similar to the approach taken by Li et al. (2018, §B.1). Finally, $f_{\text{down}}(\cdot)$ differs between the two modules, as for $r^{\text{cont}}(\cdot)$ it projects down to one dimension (ie $q = 1$) (to later go through a sigmoid function and compute a stop probability), whereas for $r^{\text{reagent}}(\cdot)$, $f_{\text{down}}(\cdot)$ projects to a dimensionality of 100 (ie $q = 100$) to form the reagent embedding.

The modules for $f^{\text{add}}(\cdot)$ and $f^{\text{remove}}(\cdot)$, that operate on each node to produce an action logit, are both NNs consisting of one hidden layer of 100 units. Concatenated onto the node features going into these networks are the node features belonging to the previous atom on the path.

The final function, $f^{\text{start}}(\cdot)$, is represented by an NN with hidden layers of 100 units. When conditioning on reagents (ie for ELECTRO) the reagent embeddings calculated by $r^{\text{reagent}}(\cdot)$ are concatenated onto the node embeddings and we use two hidden layers for our NN. When ignoring reagents (ie

for ELECTRO-LITE) we use one hidden layer for this network. In total ELECTRO has approximately 250,000 parameters and ELECTRO-LITE has approximately 190,000.

Although we found choosing the first entry in the electron path is often the most challenging decision, and greatly benefits from reagent information, we also considered a version of ELECTRO where we fed in the reagent information at every step. In other words, the modules for $f^{\text{add}}(\cdot)$ and $f^{\text{remove}}(\cdot)$ also received the reagent embeddings calculated by $r^{\text{reagent}}(\cdot)$ concatenated onto their inputs. On the mechanism prediction task (Table 2) this gets a slightly improved top-1 accuracy of 78.4% (77.8% before) but a similar top-5 accuracy of 94.6% (94.7% before). On the reaction product prediction task (Table 3) we get 87.5%, 94.4% and 96.0% top-1, 3 and 5 accuracies (87.0%, 94.5% and 95.9% before). The tradeoff is this model is somewhat more complicated and requires a greater number of parameters.

## C.2 TRAINING

We train everything using Adam (Kingma and Ba, 2015) and an initial learning rate of 0.0001, which we decay after 5 and 9 epochs by a factor of 0.1. We train for a total of 10 epochs. For training we use reaction minibatch sizes of one, although these can consist of multiple intermediate graphs.

## D FURTHER DETAILS ON IDENTIFYING REACTIONS WITH LINEAR FLOW TOPOLOGY

This section provides further details on how we extract reactions with linear electron flow topology, complementing Figure 3 in the main text. We start from the USPTO SMILES reaction string and bond changes and from this wish to find the electron path.

The first step is to look at the bond changes present in a reaction. Each atom on the ends of the path will be involved in exactly one bond change; the atoms in the middle will be involved in two. We can then line up bond change pairs so that neighboring pairs have one atom in common, with this ordering forming a path. For instance, given the pairs "11-13, 14-10, 10-13" we form the unordered path "14-10, 10-13, 13-11". If we are unable to form such a path, for instance due to two paths being present as a result of multiple reaction stages, then we discard the reaction.

For training our model we want to find the ordering of our path, so that we know in which direction the electrons flow. To do this we examine the changes of the properties of the atoms at the two ends of our path. In particular, we look at changes in charge and attached implicit hydrogen counts. The gain of negative charge (or analogously the gain of hydrogen as $H^+$ ions without changing charge) indicates that electrons have arrived at this atom, implying that this is the end of the path; vice-versa for the start of the path. However, sometimes the difference is not available in the USPTO data, as unfortunately only major products are recorded, and so details of what happens to some of the reactant molecules' atoms may be missing. In these cases we fall back to using an element's *electronegativity* to estimate the direction of our path, with more electronegative atoms attracting electrons towards them and so being at the end of the path.

The next step of filtering checks that the path alternates between add steps (+1) and remove steps (-1). This is done by analyzing and comparing the bond changes on the path in the reactant and product molecules. Reactions that involve greater than one change (for instance going from no bond between two atoms in the reactants to a double bond between the two in the products) can indicate multi-step reactions with identical paths, and so are discarded. Finally, as a last sanity check, we use RDKit to produce all the intermediate and final products induced by our path acting on the reactants, to confirm that the final product that is produced by our extracted electron path is consistent with the major product SMILES in the USPTO dataset.

## E PREDICTION USING OUR MODEL

At predict time, as discussed in the main text, we use beam search to find high probable chemically-valid paths from our model. Further details are given in Algorithm 2. For ELECTRO this operation takes 0.337s per reaction, although we do not parallelize the molecule manipulation across the different beams, and so the majority of this time (0.193s) is used within RDKit to make intermediate

---

**Algorithm 2** Predicting electron paths at test time.

---

**Input:** Molecule $\mathcal{M}_0$ (consisting of atoms $\mathcal{A}$), reagents $\mathcal{M}_e$ , beam width $K$, time steps $T^{\mathrm{max}}$

1: $\hat{\mathcal{P}} = \{(\emptyset, \log(1 - \texttt{calc\_prob\_continue}(\mathcal{M}_0)))\}$ ▷*This set will store all completed paths.*

2: $F_{\mathrm{remove}} = 1$ ▷*Remove flag*
3:
4: $\hat{\mathcal{B}} = \emptyset$. ▷*This set will store all possible open paths. Cleared at start of each timestep.*
5: **for all** $v \in \mathcal{A}$ **do**
6: $\quad \rho = (v)$
7: $\quad p_{\mathrm{path}} = \log \texttt{calc\_prob\_continue}(\mathcal{M}_0) + \log \texttt{calc\_prob\_initial}(v, \mathcal{M}_0, \mathcal{M}_e)$
8: $\quad \hat{\mathcal{B}} = \hat{\mathcal{B}} \cup \{(\rho, p_{\mathrm{path}})\}$
9: **end for**
10: $\mathcal{B}_0 = \texttt{pick\_topK\_actions}(\hat{\mathcal{B}})$ ▷*We filter down to the top K most promising actions.*
11:
12: **for** t in $(1, \ldots, T^{\mathrm{max}})$ **do**
13: $\quad \hat{\mathcal{B}} = \emptyset$
14: $\quad$ **for all** $(\rho, p_{\mathrm{path}}) \in \mathcal{B}_{t-1}$ **do**
15: $\quad\quad \mathcal{M}_\rho = \texttt{calc\_intermediate\_mol}(\mathcal{M}_0, \rho)$
16: $\quad\quad p_c = \texttt{calc\_prob\_continue}(\mathcal{M}_\rho)$
17: $\quad\quad \hat{\mathcal{P}} = \hat{\mathcal{P}} \cup \{(\rho, p_{\mathrm{path}} + \log(1 - p_c))\}$
18: $\quad\quad$ **for all** $v \in \mathcal{A}$ **do**
19: $\quad\quad\quad \rho' = \rho^\frown(v)$ ▷*New proposed path is concatenation of old path with new node.*
20: $\quad\quad\quad v_{t-1} = $ last element of $\rho$
21: $\quad\quad\quad \hat{\mathcal{B}} = \hat{\mathcal{B}} \cup \{(\rho', p_{\mathrm{path}} + \log p_c + \log \texttt{calc\_prob\_action}(v, \mathcal{M}_\rho, v_{t-1}, F_{\mathrm{remove}}))\}$
22: $\quad\quad$ **end for**
23: $\quad$ **end for**
24: $\quad \mathcal{B}_t = \texttt{pick\_topK\_actions}(\hat{\mathcal{B}})$
25: $\quad F_{\mathrm{remove}} = F_{\mathrm{remove}} + 1 \mod 2$. ▷*If on add step change to remove and vice versa.*
26: **end for**
27:
28: $\hat{\mathcal{P}} = \texttt{sort\_on\_prob}(\hat{\mathcal{P}})$

**Output:** Valid completed paths and their respective probabilities, sorted by the latter, $\hat{\mathcal{P}}$

---

molecules and extract their features. At test time we take advantage of the embarrassingly parallel nature of the task to parallelize across test inputs. To compute the log likelihood of a reaction (with access to intermediate steps) it takes ELECTRO 0.007s.

## F    FURTHER EXAMPLE OF ACTIONS PROPOSED BY OUR MODEL

This section provides further examples of the paths predicted by our model. In Figures 8 and 9, we wish to show how the model has learnt chemical trends by testing it on textbook reactions. In Figure 10 we show further examples taken from the USPTO dataset.

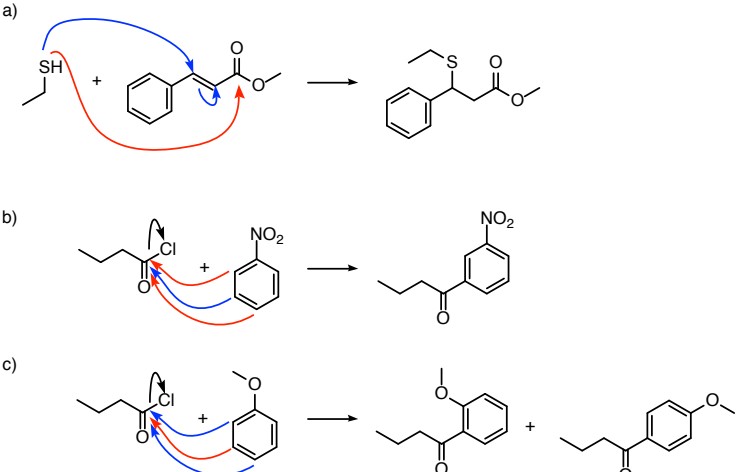

Figure 8: Predicted mechanism of our model on reactant molecules. Green arrow shows preferred mechanism, whereas pink shows the model's second preferred choice. Here, the first-choice prediction is incorrect, but chemically reasonable, as the Weinreb amide is typically used together in reactions with Magnesium species. The second-choice prediction is correct.

Figure 9: Additional typical selectivity examples: Here, the expected product is shown on the right. The blue arrows indicate the top ranked paths from our model, the red arrows indicate other possibly competing but incorrect steps, which the model does not predict to be of high probability. In all cases, our model predicted the correct products. In b) and c), our model correctly recovers the regioselectivity expected in electrophilic aromatic substitutions.

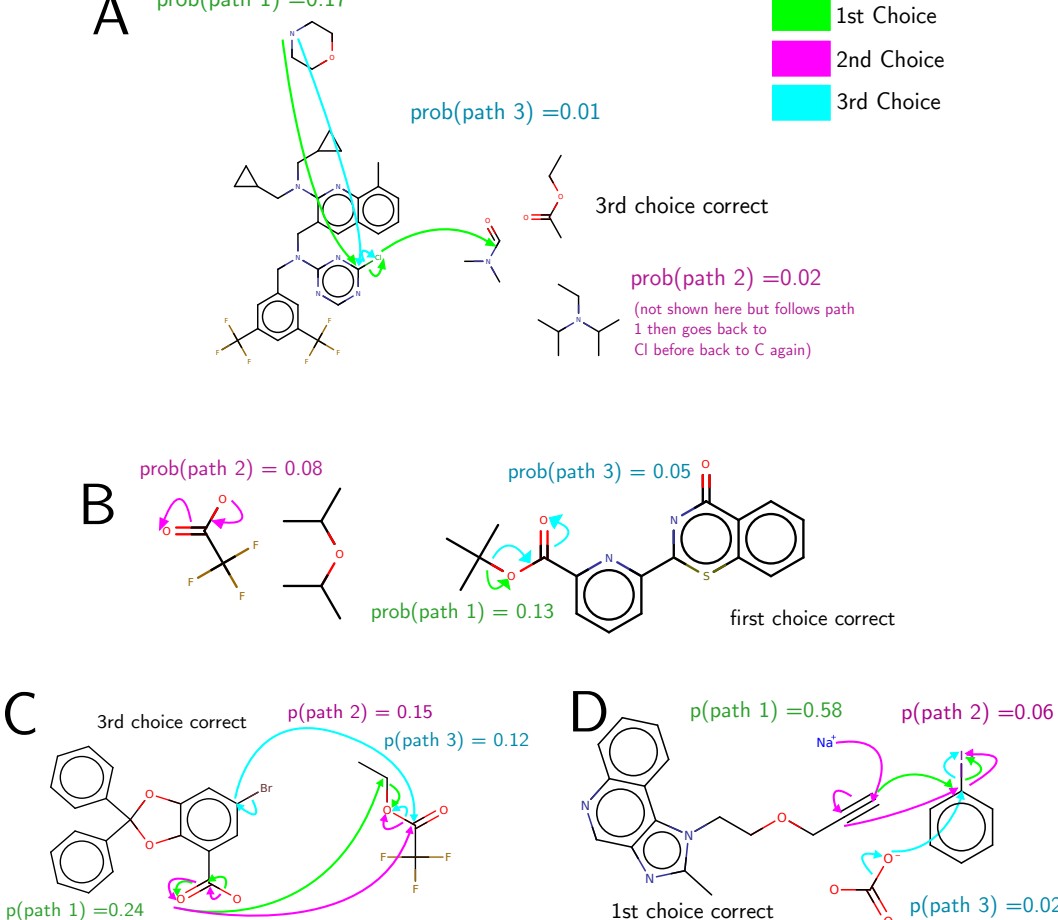

Figure 10: Four examples of the paths predicted by the ELECTRO-LITE. (These reactions have been taken from the USPTO dataset and have not been seen by the model in training).

