# OpenReview forum: "A Generative Model For Electron Paths"
_ICLR.cc/2019/Conference_

### Official Review · AnonReviewer2 · 2018-10-25
**good paper, nice contribution**

**Rating:** 8
**Confidence:** 4

**Review:**

The paper presents a novel end-to-end mechanistic generative model of electron flow in a particular type of chemical reaction (“Linear Electron Flow” reactions) . Interestingly, modeling the flow of electrons aids in the prediction of the final product of a chemical reaction over and above problems which attack this “product prediction problem” directly. The method is also shown to generalize well to held-out reactions (e.g. from a chemistry textbook).

General Impressions

+ For me the biggest selling point is that it improves performance in predicting the ultimate reaction outcome. It should do because it provides strictly more supervision, but it’s great that it actually does.
+ Because it models the reaction mechanism the model is interpretable, and it’s possible to enforce constraints, e.g. that dynamics are physically possible.
+ Generalises outside of the dataset to textbook problems :-)
+ Well-founded modeling choices and neural network architectures.
- Only applies to a very particular type of reaction (heterolytic LEF).
- Requires supervision on the level of electron paths. This seems to inhibit applying the model to more datasets or extending it to other types of reactions.
- Furthermore the supervision extraction does not seem take advantage of symmetries noted in the section(s) about difficulty evaluating inference.
- It would be nice to margin out the electron flow model and just maximize the marginal likelihood for the product prediction problem.

Novelty
I’m not an expert on the literature of applying machine learning to the problems of reaction {product, mechanism} prediction but the paper appears to conduct a thorough review of the relevant methods and occupy new territory in terms of the modeling strategy while improving over SOTA performance.

Clarity
The writing/exposition is in general extremely clear. Nicely done. There are some suggestions/questions which I think if addressed would improve clarity.

Ways to improve the paper
1. Better motivate the use of machine learning on this problem. What are the limitations of the arrow-pushing models?

2. Explain more about the Linear Electron Flow reactions, especially:
- Why does the work only consider “heterolytic” LEF reactions, what other types of LEF reactions are omitted?
- Is the main blocker to extending the model on the modeling front or the difficulties of extracting ground-truth targets? It appears to be the latter but this could be made more clear. Also that seems to be a pretty severe limitation to making the algorithm more general. Could you comment on this?

Questions
1. Is splitting up the electron movement model into bond “removal” and “addition” steps just a matter of parameterization or is that physically how the movements work?

2. It appears that Jin et al reports Top 6/8/10 whereas this work reports Top 1/3/5 accuracy on the USPTO dataset. It would be nice if there was overlap :-). Do your Top 6/8/10 results with the WLDN model agree with the Jin et al paper?


Nits
Section 2.3, first paragraph “...(LEF) topology is by far the most important”: Could you briefly say why? It’s already noted that they’re the most common in the database. Why?

Section 3.ElectionMovement, first paragraph. “Observer that since LEF reactions are a single path of electrons…”. Actually, it’s not super clear what this means from the brief description of LEF. Can you explain these reactions in slightly more detail?

Section 3.ElectionMovement, second paragraph. “Differently, the above distribution can be split…”. Awkward phrasing. How about “In contrast, the above distribution can be split…”.

Section 3.Training, last sentence “...minibatches of size one reaction”. Slightly awkward phrasing. Maybe “...minibatches consisting of a single reaction”?

Section 5.2, second sentence. “However, underestimates the model’s actual predictive accuracy…”. It looks like a word accidentally got deleted here or something.

Section 5.2, paragraph 4. “To evaluate if our model predicts the same major project”... Did you mean “the same major product”?

---

> ### Comment · AnonReviewer3 · 2018-11-02
> **impossible to tell if "improves performance" without some variance measure**
>
> In response to "the biggest selling point is that it improves performance in predicting the ultimate reaction outcome" -- in fact it is impossible to tell if there is any significant improvement in prediction accuracy, because the paper reports no measure of variance (confidence interval or standard deviation). The results section needs to be improved by adding these details, (use K-fold cross-validation and compute mean/sd of prediction accuracy over the K test sets) so the reader can indeed determine whether or not there is any significant difference in prediction accuracy.

---

> > ### Author Response · Authors · 2018-11-22
> > **We have run K-fold CV with ELECTRO-LITE and provide further details on a comment on AnonReviewer3's initial review**
> >
> > We have responded to this comment on our answer to your review and provide a detailed response there. We hope this addresses your concerns. In summary, we show on ELECTRO-LITE using K-fold cross validation that the variation within a method is on the order of tenths of a percent, whereas the difference between the baseline methods we analyse is much greater than this.

---

> ### Author Response · Authors · 2018-11-22
> **Thank you for your review, in answer to your questions (part 1)**
>
> Thank you for your thoughtful and encouraging review! We go through your comments and questions below.
>
> ## Marginalisation and Symmetries
> Thank you for your suggestion to marginalise out over the different paths when optimizing solely for reaction prediction. We think this is an interesting suggestion and we would like to explore this in future work. A challenge remains on how best to perform the marginalisation as the action space can be very large, so we would probably have to sample.
>
> Although we do not account for symmetries when extracting a supervision signal, as we are using graph neural networks to parameterize our functions, the probabilities we predict for the two (or more) symmetrical actions should be equal. Constructing the automorphism group of a graph is computationally at least as hard as the graph isomorphism problem so we have avoided doing this during training so far, however we agree it would make for interesting future work.
>
>
> ## Improvements
> Thank you for the suggested improvements. We will add a discussion on these points to our paper, as we agree that it will improve the paper. We also briefly discuss these points below.
>
> ### Improvements 1. Motivate machine learning approach to learn arrow pushing models. Limitations to arrow-pushing models
>
> Arrow pushing models (and more generally methods that model molecules as graphs) abstract away the details about the electronic structure, and conformational information, ie information about how the molecule shape changes in 3D. This information is crucial in some cases. That said the arrow-pushing abstraction is extremely powerful. It allows chemists to make very quick, but accurate predictions without doing any quantum simulations, just using pencil and paper, and to understand relations between reaction classes, which is often not possible using quantum mechanics alone.
>
> We believe that using ML to learn arrow pushing models is a sensible and beneficial approach. Currently this task is done by expert chemists. We are building off a simplification and abstraction that chemists have shown to be useful and powerful. Using ML to learn reactions in this way makes our model interpretable and easy to query.
>
> ###  Improvements 2.  LEF reactions
> > Why only ‘heterolytic’ LEF reactions? Are there other types of LEF reactions?
> There are also ‘homolytic’ LEF reactions which involve a single electron moving (instead of a pair). We will clarify this in the manuscript! However, we will leave their treatment for future work.
>
> > Challenges of extending the model on the modelling front versus the data collection front?
> Yes we believe that the model could be extended in future work to deal with a greater class of reactions. For instance some reactions can be broken down into multiple electron paths, where several pairs of electrons get shifted at the same time, which could be modelled by simply running ELECTRO multiple times. However, yes extracting paths to train on (if they overlap) remains an outstanding challenge, perhaps requiring some human supervision or quantum mechanical calculations for creating any training set. Also, having access to more fine-grained datasets, which not only feature the reactants and products of reactions, but also identifiable (stable) intermediates, would likely allow better predictions.
> Having said that, the LEF reactions that ELECTRO currently can handle are very common (they make up over 70% of the reactions in the USPTO dataset) and we hope that the heuristics and trends the model learns on this set will also be of use when making predictions for other reaction types.

---

> > ### Author Response · Authors · 2018-11-22
> > **Thank you for your review, in answer to your questions (part 2)**
> >
> > (previous comment continued to answer the remaining questions)
> >
> > ## Questions
> >
> > ### Questions 1. Is splitting up the electron movement model into bond “removal” and “addition” steps just a matter of parameterization or is that physically how the movements work?
> >
> > It’s physically how the movements work, the LEF class of reactions consists of movements which effectively remove and then add bonds. However, we note that representing the molecules as graphs is an abstraction, although powerful, and there are subtleties not contained in our model, such as conformational information.
> >
> > ### Questions 2. It appears that Jin et al reports Top 6/8/10 whereas this work reports Top 1/3/5 accuracy on the USPTO dataset. It would be nice if there was overlap :-). Do your Top 6/8/10 results with the WLDN model agree with the Jin et al paper?
> >
> > It looks like we both report top 1/3/5 accuracy but that perhaps confusion is arising from comparing different tables?
> >
> > One can think of the Jin et al model consisting of two parts: (1) The reaction centre predictor, which generates pairs of atoms for which bonds may change (2) The candidate ranker, which evaluates enumerated configuration changes between the pairs. This second stage comes up with the final product.
> >
> > In table 1a Jin et al report the coverage of the true reaction bonds when including more reaction pairs. This they do after filtering for the top 6/8/10 candidates, and this is perhaps what you are referring to? We do not have the same pipeline of filter, enumerate, rank and so we cannot (and it does not make sense for us) to run this experiment.
> >
> > In table 1b they report top 1/3/5 accuracy for the reaction prediction task (given that stage 1 is fixed to give 6 pairs). And hence we use the same accuracies in this work.
> >
> > Hope this clears up the confusion :).
> >
> > ## Nits
> > Thanks for picking up these typos and other areas for small improvement. We shall fix/describe these things!

---

### Official Review · AnonReviewer3 · 2018-11-02
**Potentially interesting and novel ideas but impossible to tell if they are significant due to low-quality results section**

**Rating:** 4
**Confidence:** 4

**Review:**

Review of "A Generative Model for Electron Paths"

Paper summary:

The paper proposes a new model for predicting arrow-pushing chemical
reaction diagrams from raw reaction data.

Section 1 summarizes the motivation: whereas other models only predict
chemical reaction products from reactants, the proposed model attempts
to also predict the reaction mechanism.

Section 2 provides a background on related work. Previous models for
mechanism prediction are limited to work which require expert-curated
training sets. The proposed model is designed for a subset of
reactions called "linear electron flow" (LEF) which is
explained. Contributions of this paper are an end-to-end model, a
technique for identifying LEF reactions/mechanisms from
reaction/product data, and an empirical study of how the model learns
chemical knowledge.

Section 3 explains the proposed generative model, which represents a molecule
using a graph (nodes are atoms and edges are bonds). It is proposed to
learn a series of electron actions that transform the reactants into
the products. The total probability is factorized into three parts:
starting location, electron movement, and reaction
continuation. Figure 2 and Algorithm 1 are helpful.

Section 4 explains the proposed method for creating mechanism data
from chemical reactant/product databases. Figure 3 is helpful.

Section 5 discusses results of predicting mechanisms and products on
the USPTO data set.

Comments:

Strong points of the paper: (1) it is very well written and easy to
understand, (2) the chemical figures are very well done and helpful,
and (3) the method for predicting mechanisms seems to be new.

The major weak point of the paper is the results section, which needs
to be improved before publication.

In particular Tables 2-3 (comparison of prediction accuracy) need to
show some measure of variance (standard deviation or confidence
interval) so the reader can judge if there is any significant
difference between models. Please use K-fold cross-validation, and
report mean/sd of test accuracy over the K test folds.

The term "end-to-end" should be defined. In section 2.2 it is written
"End-to-End: There are many complex chemical constraints that limit
the space of all possible reactions. How can we differentiate through
a model subject to these constraints?" which should be clarified using
an explicit definition of "end-to-end."

Also there needs to be some comparison with baseline methods for
predicting mechanisms.  It is claimed that no comparison can be made
against the previous methods for mechanism prediction (Section 2.2),
because "they require expert-curated training sets, for which organic
chemists have to hand-code every electron pushing step." However the
current paper proposes a method for generating such steps/data for LEF
reactions. So why not use those data to train those baseline models,
and compare with them? That would make for a much stronger paper. Please
add at least one of the methods discussed in section 2.2 to your
accuracy comparison in Table 2.

It would additionally be helpful to know what the "uninformed
baseline" / "ignore the inputs" / "random guessing" accuracy rates are
on your data set. For example in classification the uninformed
baseline always predicts the class which is most frequent in the
training data, and in regression it predicts the mean of the
labels/outputs in the training data. What would the analogy be for
your two problems? (product and mechanism prediction)

---

> ### Comment · AnonReviewer1 · 2018-11-03
> **practical restrictions?**
>
> Your point is well taken and I personally also think this tradition can be problematic, but probably due to the practical computational cost, evaluations on a single shot training/test split would be standard and often considered as acceptable (for example, consider ImageNet cases) when we use quite complicated neural networks trained with large datasets. We can't get SDs/CIs from one-shot evaluations...
>
> USPTO seems also quite a large dataset each representing a set of graphs, and even for LEF reactions (349,898 reactions, of which 29,360 form the held-out test set as in p.7).

---

> ### Author Response · Authors · 2018-11-22
> **Thank you for your review, in answer to your questions (part 1)**
>
> Thank you for taking the time to review our paper. We go through your concerns in more details below. As a brief summary we:
> Show on ELECTRO-LITE using K-fold cross validation that the variation within a method is on the order of tenths of a percent, whereas the difference between the baseline methods we analyse is much greater than this.
> Explain how we can encode chemical restraints whilst being able to compute the gradient of the parameters of our end-to-end model
> Provide the results for a random baseline on the mechanism prediction task. The top-1 accuracy of such a baseline would be less than one percent, due to the large number of possible actions at each step.
>
>
> ## K-fold cross validation
> While we agree that K-fold cross validation would be ideal, practically, as mentioned by the other reviewer’s comment, it would be difficult; it would require considerable computational resources. This is particularly true for the seq2seq model, which is compute hungry while training (to be fair to other methods we would need to cross validate these methods too, as this has not been done by these works).
>
> We also wish to make it clear that:
> We used the relevant LEF subsets of the same pre-defined train, validation and test sets used by Jin et al  (2017) and Schwaller et al (2018).
> This methodology, following the common task framework (Donoho D (2015), Section 6), of developing and testing on pre-defined splits of a dataset is used throughout ML from machine translation to image classification, such as when evaluating models using the ML benchmarks: ImageNet, CIFAR-10 or even MNIST.
> All development of the algorithm was done using only the training and the validation sets, with the test set only being used for the final evaluation to get the numbers reported in this paper.
>
> Note that we are using a large test set of 29360 items.Treating the probability of success on each test set item as a Bernoulli variable, our top-1 reported mean product prediction accuracies for ELECTRO and ELECTRO-LITE have standard deviations less than 0.25% (0.24% for ELECTRO-LITE and 0.20% for ELECTRO).
>
> Furthermore, we also tried to rule out that significant variance could occur due to training/test set differences by performing 3-fold cross validation with the ELECTRO-LITE model. In order to do this we first merged the current training, validation and test sets. For the mechanism prediction task (Table 2) we report the results of these runs in the Table below:
>
>                          Accuracies (%)
>                  top-1    |  top-3   |   top-5
> ------------------------------------------------
> Fold 1    |   69.8     |  87.2    |   91.6
> Fold 2    |   70.0     |  87.2    |   91.5
> Fold 3    |   69.5     |  87.0    |   91.3
>
> Note that these figures are not directly comparable to the previous work as the training/test set sizes have changed. In particular, the training set has got smaller and the test set larger. However, this suggests the variation is on the order of tenths of a percent, whereas the difference between the different methods we analyse is much greater than this.
>
>
> Donoho D (2015) 50 years of Data Science. URL http://courses. csail. mit. edu/18 337: 2015.
>
> (comment continued separately due to space)

---

> > ### Author Response · Authors · 2018-11-22
> > **Thank you for your review, in answer to your questions (part 2)**
> >
> > (comment continued)
> >
> > ## End-to-end models and how we can differentiate through our model end-to-end subject to chemical constraints
> > By end-to-end we mean that our full model can be trained from input to output purely using gradient-based techniques. In our approach this manifests itself by training on each action of the model simultaneously. At train time this is possible by conditioning on the correct previous actions from previous time steps when predicting the actions at latter time steps.
> >
> > This contrasts with previous approaches to this problem (e.g. Jin et al., 2017 or Kayala & Baldi, 2011, 2012) which break down then problem into several steps and separate models. These models have to be trained in stages. These approaches often can be broken down into three stages (i) ML based filtering of ‘reaction sites’ (ii) manual enumeration of all possible changes that can occur at these ‘reaction sites’, (iii) ML based ranking of these enumerated options. As these approaches split the problem down into separate processes, they cannot leverage the power of state-of-the-art gradient-based techniques to solve the full problem. This means their solutions are likely suboptimal local minima when composed. On the other hand, our model adjusts all parameters simultaneously to solve the goal of reaction prediction. As shown in our experimental results, this end-to-end approach allows us to improve upon prior work, with the added benefit of approaching the problem in a chemist-interopratible way.
> >
> > The chemical constraints are encoded as masked out operations in our model. These disallowed operations are shown as red crosses in Figure 2 and are represented by the $\beta$ terms in eqns 2 through 4. Note that we never have to differentiate through a masked out operation during training, as by definition these do not happen, thus our model can still be end-to-end.
> >
> >
> > ## Mechanism Prediction Baseline
> > Sorry for the confusion here. The previous mechanism prediction work has used a private, expert-curated training set. These datasets include expert information about electron sources and sinks as well as reaction conditions such as temperature and anion/action solvation potential (Kayala and Baldi, 2011; Section 2). This has meant these datasets are often small (Fooshee et al, 2018 (Section 2.3) has a dataset size of around 11 000).
> >
> > You are correct, we could use our approximate reaction mechanism extraction method to label sources and sinks. However, we would still not be able to provide the full reaction conditions data, as this data does not currently exist in the patent dataset (Lowe, 2012, Section 4.11.8).
> >
> > Moreover, a separate issue is that these previous mechanism methods also need expert-curated features. This includes molecular orbital data and steric information among others. These features are hard to encode, indeed in the earlier work and until  Fooshee et al (2018) the chemical model used for their reaction predictor could not handle the elements  Sc, Ti, Zn, As, or Se.  As well as requiring these features, extra expert-encoded constraints are required, such as the number of bonds particular elements can form.
> >
> > It is due to these requirements that we have described these methods as needing ‘expert-curated’’ training sets. However, we shall make this clearer in our paper.
> >
> > This so far has described why we cannot run their methods on our dataset. Alternatively, we cannot run our method on their dataset as their data is currently private (they were also unable to release it via email).
> >
> > We think the reaction predictor proposed in these previous works is an interesting model and are disappointed that we are unable to compare against it currently on any benchmark task. We hope to open source our code in the future so that comparisons to ELECTRO can be made by others.
> >
> >
> > ## ‘Random guessing’ accuracies on our dataset
> > It is easiest to compare against a random baseline on the reaction mechanism task where the exact steps, including how many there are, is known. We consider a random guessing model, which assigns equal weight to each atom. However, we keep the masking we use with ELECTRO so that the random model is restricted to chemically plausible options.
> > On the 29360 test set the random baseline would get the correct answer of a reaction with a mean probability of 5x10^-5. This is exceedingly low and is due to the very large number of actions the model can take on the initial select and add steps.

---

### Official Review · AnonReviewer1 · 2018-11-03
**A quite interesting contribution that also brings more clearer interpretations on what is learned**

**Rating:** 8
**Confidence:** 4

**Review:**

Summary:
The paper presents a novel method for predicting organic chemical reactions, in particular, for learning (Robinson-Ingold's) ''arrow pushing" mechanisms in an end-to-end manner. Organic molecules consist of covalent bonds (that's why we can model them as molecular graphs), and organic reactions are recombinations of these bonds. As seen in organic chemistry textbooks, traditional chemists would qualitatively understand organic reactions as an alternating series of electron movements by bond breaking (bond cleavage) and bond forming (bond formation). Though now quantum chemistry calculations can give accurate quantitative predictions, these qualitative understanding of organic reactions still also gives strong foundations to consider and develop organic reactions. The proposed method tries to learn these series of bond changes directly through differentiable architectures consisting of three graph neural networks: 1) the one for determining the initial atom where electron movements start, 2) the one for representing state transitions from  the previous bond change to the next, and 3) the one for determining when the electron movements end. Experimental evaluations illustrate the quantitative improvement in final product prediction against existing methods, as well as give chemical intuitions that the proposed method can detect a class of LEFs (linear electron flows).

Comment:
- This study is a quite interesting contribution because many existing efforts focus on improving differentiable architecture design for graph transformation and test it using chemical reaction data without considering what is learned after all. In contrast, this paper gives the clear meaning to predict "arrow pushing" mechanism from chemical reaction data and also makes sure the limitation to LEFs that are heterolytic. Traditional graph rewrite systems or some recent methods directly borrowing ideas from NLP do not always give such clear interpretations even though it can somehow predict some outputs.

- The presentation of the paper is clear and in very details, and also provides intuitive illustrative examples, and appendix details on data, implementations, and related knowledge.

- The architecture is based on graph neural networks, and seem natural enough. Basically, I liked overall ideas and quite enjoyed them but several points also remained unclear though I'm not sure at all about chemical points of view.

1) the state transition by eq (2)-(4) seems to assume 1-st order Markovian, but the electron flow can have longer dependence intuitively. Any hidden states are not considered and shared between these networks, but is this OK with the original chemical motivations to somehow model electron movements? The proposed masking heuristics to prevent stalling would be enough practically...? (LEF limitations might come from this or not...?)

2) One thing that confuses me is the difference from approaches a couple of work described at the beginning of section 'Mechanism prediction (p.3)', i.e. Fooshee et al 2018; Kayala and Baldi, 2011, 2012; Kayala et al, 2011. I don't know much about these studies, but the paper describes as "they require expert-curated training sets, for which organic chemists have to hand-code every electron pushing step". But for "Training" (p.6) of the proposed method, it also describes "this is evaluated by using a known electron path and intermediate products extracted from training data". Does this also mean that the proposed method also needs a correct arrow pushing annotations for supervised learning?? Sounds a bit contradicting statements?

3) Is it just for computational efficiency why we need to separate reactants and reagents? The reagent info M_e is only used for the network for "starting location", but it can affect any intermediate step of elementary transitions intuitively (to break through the highest energy barrier at some point of elementary transitions?). Don't we need to also pass M_e to other networks, in particular, the one for "electron movement"?

---

> ### Author Response · Authors · 2018-11-22
> **Thank you for your review, in answer to your questions**
>
> Thank you for your thoughtful and encouraging review! We go through your questions below.
>
> ## 1. 1st order Markovian
> Although the quantum-mechanical reaction mechanism is Markov, it’s true that this does not necessarily hold when considering models with a graph state, as the graph representation does not fully capture all the details of the electronic structure in some corner cases (see e.g. the textbook Gasteiger - Chemoinformatics). However, we believe the graph abstraction of molecules and their reactions to be a powerful and useful representation, due to its employment in previous machine learning approaches (eg Jin et al, 2017) and its widespread use by chemists. Therefore, we believe using a Markovian model on the molecular graph is a sensible assumption and one that is validated by our strong results. In practice we do not notice ELECTRO undoing its own work or stalling.
>
> We agree that an exciting future direction is to extend the model to cover a greater class of reactions. For this, exploring the non-Markovian structure you describe in a chemically-reasonable model would definitely be a sensible thing to do.
>
>
> ## 2. Difference to previous mechanism prediction work
> Sorry for the confusion here. The previous mechanism prediction work has used a private, expert-curated training set. These datasets include expert-curated information about electron sources and sinks as well as reaction conditions such as temperature and anion/action solvation potential (Kayala and Baldi, 2011; Section 2). This has meant these datasets are often small (Fooshee et al, 2018 (Section 2.3) has a dataset size of around 11 000).
>
> You are correct, we could use our approximate reaction mechanism extraction method to label sources and sinks. However, we would still not be able to provide the full reaction conditions data, as this data does not currently exist in the patent dataset (Lowe, 2012, Section 4.11.8).
>
> Moreover, a separate issue is that these previous mechanism methods also need expert-defined features. These features include molecular orbital data and steric information among others. These features are hard to encode, indeed in the earlier work and until Fooshee et al (2018) the chemical model used for their reaction predictor could not handle the elements  Sc, Ti, Zn, As, or Se.  As well as requiring these features, extra expert-encoded constraints are required, such as the number of bonds particular elements can form.
>
> It is due to these requirements that we have described these methods as needing ‘expert-curated’’ training sets. However, we shall make this clearer in our paper.
>
>
> ## 3. Why reagents are not passed in at later steps.
> It is indeed largely for computational reasons that we separate reactants and reagents, and pass reagents to just the starting network. We found that choosing the first entry in the electron path is often the most challenging decision, and that action steps after this have access to the previous atom as context, making it an easier task.
> Qualitatively, when running Electro-lite on the separate validation set we would see that the model often had the most errors on the first step, and that after picking this first step the next stages would often be correctly predicted.
> We have tested a version of Electro where reagent information is fed in as context at each step.
> On the mechanism prediction task (Table 2) this gets a slightly improved top-1 accuracy of 78.4% (77.8% before) but a similar top-5 accuracy of 94.6% (94.7% before). On the reaction product prediction task (Table 3) we get 87.5%, 94.4% and 96.0% top-1, 3 and 5 accuracies (87.0%, 94.5% and 95.9% before). The tradeoff is this model is somewhat more complicated and requires a greater number of parameters.

---

### Meta-Review · Area_Chair1 · 2018-12-14
**Interesting application of graph neural networks**

**Confidence:** 3
**Recommendation:** Accept (Poster)

**Metareview:**

The paper presents a graph neural network that represents the movements of electrons during chemical reactions, trained from a dataset to predict reactions outcomes.

The paper is clearly written, the comparisons are sensical. There are some concerns by reviewer 3 about the experimental results: in particular the lack of a simpler baseline, and the experimental variance. I think the some of the important concerns from reviewer 3 were addressed in the rebuttal, and I hope the authors will update the manuscript accordingly.

Overall, this is fitting for publication at ICLR 2019.